# Dietary Zinc Is Associated with Cardiac Function in the Older Adult Population

**DOI:** 10.3390/antiox12020265

**Published:** 2023-01-24

**Authors:** Iwona Szadkowska, Tomasz Kostka, Rafał Nikodem Wlazeł, Łukasz Kroc, Anna Jegier, Agnieszka Guligowska

**Affiliations:** 1Department of Sports Medicine, Medical University of Lodz, Pomorska 251, 92-213 Lodz, Poland; 2Department of Geriatrics, Healthy Ageing Research Centre, Medical University of Lodz, Plac Hallera 1, 90-647 Lodz, Poland; 3Department of Laboratory Diagnostics and Clinical Biochemistry, Medical University of Lodz, Pomorska 251, 92-213 Lodz, Poland

**Keywords:** dietary antioxidants, dietary zinc intake, cardiovascular health, aging, echocardiography, ejection fraction

## Abstract

The elderly is a group at particularly high cardiovascular risk. The coexistence of chronic diseases and use of multiple medications creates the need to look for non-pharmacological agents to improve cardiovascular health in that population. In view of reports on the potential role of zinc in enhancing pathways of myocardial tissue repair, the aim of this study was to evaluate the association between dietary zinc intake and cardiac structure and function in individuals of advanced age. The study group included 251 community-dwelling patients, with a median age of 80 years. Dieta 6.0 software was used for calculation of zinc consumption. Percentage of Recommended Dietary Allowance (RDA) for zinc correlated with left ventricular ejection fraction (LVEF) (r = 0.196, *p* < 0.05), left ventricular mass index (r = −0.137, *p* < 0.05) and tricuspid annular plane systolic excursion (TAPSE) (r = 0.153, *p* < 0.05), while zinc density did so with E/E’ ratio (r = −0.127, *p* < 0.05). In a multiple stepwise regression analysis, the best determinants of LVEF were %RDA for zinc (*p* = 0.014; β = 0.143), presence of coronary artery disease (*p* < 0.001; β = −0.39) and age (*p* = 0.036; β = −0.12). Furthermore, %RDA for zinc (*p* = 0.009; β = 0.16), female sex (*p* = 0.005; β = −0.171), beta-blocker use (*p* = 0.024; β = −0.136), body mass index (*p* = 0.008; β = 0.16) and heart rate (*p* = 0.0006; β = −0.209) had an independent effect on TAPSE. In conclusion, in individuals of very advanced age, lower zinc intake is associated with poorer cardiac function. Therefore, increasing the recommended zinc intake in this group deserves consideration.

## 1. Introduction

Cardiovascular diseases are widely spread in the general population, being not only the leading cause of death in developed countries but also generating high healthcare-related costs. The prevalence and incidence of hypertension, coronary artery disease or heart failure have been rising, especially in the older population [1,2,3].

Advanced age is one of the key non-modifiable risk factors of adverse cardiovascular events. Apart from pharmacotherapy, guidelines on cardiovascular disease prevention emphasize non-pharmacological management aimed at improving general health [4]. Along with other management approaches, adequate nutrition is important in both primary and secondary prevention of chronic diseases, including in the oldest population.

A proper diet not only provides the necessary nutrients and energy but also constitutes the most important source of antioxidants. There are numerous reports on omega-3 fatty acids, polyphenols, vitamins and their effect on the cardiovascular system, while zinc is one of the most important minerals with antioxidant properties. In developed countries, the microelement often fails to receive much attention as severe zinc deficiencies are rare. However, it is well-known that its deficiency may play an important role in the aging process and in the etiology of many chronic diseases, such as neurodegenerative diseases, immunosenescence, atherosclerosis, or cancer [5]. It has also been reported that low zinc intake in men is associated with higher mortality from coronary artery disease [6]. Zinc is found in active centers of many enzymes and proteins and ensures the efficiency of the immune system. It is also essential for DNA synthesis, RNA transcription, cell division and activation [7].

The role of zinc as an essential microelement that affects myocardial and vascular properties in multiple physiologic pathways has been previously emphasized [8,9,10,11]. However, the clinical significance of these processes is still unclear and widely discussed. Regarding the heart muscle, studies report an association between zinc deficiency and systolic or diastolic dysfunction; however, they mainly focus on patients with significantly impaired cardiac function [12,13,14,15,16]. Therefore, it is important to analyze the body’s zinc status, which can be assessed by various methods, e.g., plasma levels of zinc, the activity of zinc-containing enzymes, and dietary zinc intake [17].

The elderly is a group at particularly high cardiovascular risk. The coexistence of multiple chronic diseases and polypharmacy creates the need to look for non-pharmacological agents to improve cardiovascular health in that population. In view of reports on a potential role of zinc in enhancing pathways of myocardial tissue repair, the aim of our study was to evaluate the association between dietary zinc intake and parameters of cardiac function in individuals of advanced age.

## 2. Materials and Methods

### 2.1. Study Population

The study population included 270 community-dwelling individuals between the ages of 77 and 91, patients of an outpatient geriatric clinic.

Initially, patients older than 80 years were targeted and referred for echocardiographic assessment. Since in this age group there were too few individuals without exclusion criteria, ultimately, patients four years younger were also allowed. The exclusion criteria were acute or uncontrolled chronic inflammatory process, end-stage chronic or neoplastic disease, hemodialysis therapy, mental or functional inability to participate in the study, the use of zinc supplementation and lack of patient consent. All the patients underwent clinical assessment including a review of medical history and basic demographic information, physical examination, dietary assessment and echocardiography. Following an evaluation of cardiac parameters, 19 patients with atrial fibrillation present on examination were excluded from further analysis.

After taking accurate anthropometric measurements—body mass, height, waist circumference assessed with the RADWAG personal scale weight (WPT60 150OW; Radwag Balances and Scales, Radom, Poland) and SECA measuring tape (SECA Deutschland, Hamburg, Germany)—the following indices were calculated: body mass index (BMI) and the waist-to-height ratio (WHtR). The level of education is presented as the number of years of education.

The study was approved by the Bioethics Committee of the Medical University of Lodz in accordance with the Declaration of Helsinki (RNN/345/08/KB). All the patients gave their written consent for participation in the study.

### 2.2. Zinc Intake

The level of zinc consumption was assessed using a modified 24 hour interview method [18,19]. To decrease the risk of errors from dietary recall, the participants were instructed to prepare a list of food products (including snacks and beverages) they ate on the day before the appointment. Since the diet of older people is monotonous, with no major modifications throughout the week, this method is considered to be the most effective one in the older population [20,21]. The interview was conducted professionally—without any judgment or comments—by experts (dieticians and nutritionists) so as not to exert any impact on the participant. The interviewers also asked the patients about their mealtimes, method of preparation and product details, such as the type of bread, meat and cheese.

Zinc consumption in milligrams was calculated with the use of Diet 6.0 software (National Food and Nutrition Institute, Warsaw, Poland), based on the Polish base of the composition and nutritional value of food [22,23]. The software also compared intake levels with the body’s zinc requirements, expressed as a percentage of the Polish Recommended Dietary Allowance (RDA) [24]. Zinc density was calculated as milligrams of zinc per 1000 kcal of dietary energy.

### 2.3. Echocardiography

Transthoracic echocardiography was performed by the same investigator using a Vivid S70 ultrasound system (GE Medical Systems, 2018). The parameters of cardiac structure and function were assessed according to the guidelines of the American Society of Echocardiography and the European Association of Cardiovascular Imaging [25,26]. Internal cardiac dimensions were obtained with two-dimensional linear measurements. Left ventricular (LV) end-diastolic wall thickness, LV end-diastolic dimension, left atrial (LA) anteroposterior dimension and proximal right ventricular (RV) outflow tract (RVOT) dimension were assessed from the parasternal long-axis view. Relative wall thickness (RWT) was calculated according to the following formula: 2 × diastolic posterior wall thickness divided by LV end-diastolic dimension. Left ventricular mass (LV mass) was calculated using the Devereux formula and indexed to body surface area (LV mass index—LVMI). Left ventricular end-diastolic volume (LVEDV), LV end-systolic volume (LVESV) and LV ejection fraction (LVEF) were determined using the modified biplane Simpson’s method from the apical four- and two-chamber views. LVEDV and LVESV were indexed to body surface area—LVEDV index (LVEDVI) and LVESV index (LVESVI). Left atrial volume was measured using the biplane area-length method and divided by body surface area (LA volume index—LAVI). The peak early (E), late-atrial (A) diastolic velocities were assessed by pulsed-wave Doppler from mitral inflow profile and E/A ratio was calculated. Pulsed tissue Doppler imaging (TDI) was used to obtain the LV peak systolic velocity (S’) and early diastolic velocity (E’), as mean values from the median and lateral mitral annular velocities. The average mitral E/E’ ratio was calculated. Assessment of right ventricle (RV) function included tricuspid annular plane systolic excursion (TAPSE) measured using 2D-guided M-mode and TDI peak systolic velocity (S’ RV) of the lateral tricuspid valve annulus. Reduced TAPSE is defined as below 17 mm [26]. Tricuspid regurgitation velocity (TR Vmax) was measured with the use of continuous-wave Doppler.

### 2.4. Statistical Analysis

The normality of distribution was verified using the Shapiro–Wilk test. Variables without a normal distribution were presented as median value and quartile range (from 25% to 75% quartile), with a normal distribution as median and standard deviations. Between groups, the quantitative variables were compared using the one-way ANOVA or Mann–Whitney U test and the qualitative variables using the chi-square test or Fischer’s exact test. The correlations between the numerical data were analyzed through Spearman‘s correlation coefficient.

Stepwise multiple regressions were performed to adjust the potential relationship between selected echo results (LVEF, TAPSE, LVMI, E/E’ ratio), zinc intake and variables that were statistically significant in univariate analyzes. In accordance with the guidelines specified in the literature, sex, age, BMI, diseases (coronary artery disease, hypertension, diabetes, heart failure) and medication intake (beta-blockers, angiotensin-converting-enzyme inhibitors, angiotensin receptor blockers, mineralocorticoid receptor antagonist statins) were taken into account. Qualitative variables were included in the model as dichotomized. Statistical significance was set at *p* ≤ 0.05. The analyses were performed with the use of Statistica 13.1 (StatSoft Polska, Cracow, Poland).

## 3. Results

The study population included 251 elderly individuals who appeared to be homogenous in terms of age and anthropometric parameters. The difference in the numbers of women (*n* = 180) in relation to men (*n* = 71) reflects the demographic situation in Poland. In 2019 the sex index for seniors 75+ was 0.51 [27]. It was a group of relatively well-educated people. According to medical history, coronary artery disease was manifested in a greater number of the men, including previous myocardial infarction in the subgroup. Diagnosis of heart failure was more common among the women, and diuretics were used more often in the subgroup. There were no statistically relevant differences in the prevalence of other chronic diseases between sexes in the studied population. Resting blood pressure (BP) and heart rate were comparable in both subgroups. Dietary zinc intake was higher in the men; however, the women showed a higher level of RDA norm conformity. One-third of males and almost half of females had a zinc intake ≥ RDA. The nutritional density of zinc in the diet did not differ.

The detailed baseline characteristics of the studied population are presented in Table 1.

The echocardiographic parameters are presented in Table 2. In terms of sex, higher chamber dimensions and volumes, higher LV mass/LVMI, S’ velocity, as well as TAPSE values were found in the men. LVEF and other parameters were comparable in both men and women.

Table 3 presents the correlations between dietary zinc intake, %RDA for zinc and zinc density, age and echocardiographic parameters. Due to different levels of intake standards for men and women and differences in zinc intake in milligrams per day, a correlation analysis was performed by sex, whereas analyses of %RDA for were performed for the whole group.

As regards the whole population, %RDA for zinc intake was significantly positively correlated with LVEF (r = 0.196, *p* < 0.05), and TAPSE (r = 0.153, *p* < 0.05). Moreover, negative correlation was observed with LVMI (r = −0.137, *p* < 0.05). In the sex-based analysis, there was a significant association between zinc intake—expressed as both %RDA and milligrams—and LVEF in women (r = 0.184, *p* < 0.05), and TAPSE in men (r = 0.353, *p* < 0.05). These relationships are presented in Figure 1 and Figure 2. There was no correlation between %RDA for zinc with the other echocardiographic parameters.

Age in the whole group, and in women, negatively correlated with zinc nutritional density, while positively with LAVI, E/E’ ratio; only in the total group did it correlate negatively with LVEF and positively with LVMI.

The results of the performed multiple stepwise regression indicate that zinc analyzed as %RDA (*p* = 0.014; β = 0.143), presence of coronary artery disease (*p* < 0.001; β = −0.39) and age (*p* = 0.036; β = −0.12) were the best determinants of LVEF as a dependent variable (R2 = 0.19; *p* < 0.001). Presence of diabetes or hypertension, medications used (beta-blockers, ACEI/ARB, MRA, statins), BMI and heart rate were not significant in the univariate analyses; they were not included in the multiple regression.

However, in the case of TAPSE, both BMI and heart rate showed a statistically significant correlation (r = 0.151 and r = −0.172, respectively) and in one-way ANOVA, sex, statins and beta-blockers were statistically important. All the variables were entered into the regression model. Ultimately, %RDA for zinc (*p* = 0.009; β = 0.16), female sex (*p* = 0.005; β = −0.171), beta-blocker use (*p* = 0.024; β = −0.136), BMI (*p* = 0.008; β = 0.16) and heart rate (*p* = 0.0006; β = −0.209) had a statistically significant effect on TAPSE (R2 = 0.14; *p* < 0.001).

The results of the performed multiple stepwise regression indicate that only age (beta = 0.17; *p* = 0.008) and presence of hypertension (beta = 0.16; *p* = 0.01) determinate the E/E’ ratio as a dependent variable (R2 = 0.05; *p* = 0.002). Sex, zinc intake expressed as %RDA, BMI, presence of coronary artery disease, diabetes or previous MI did not affect the E/E’ ratio.

Furthermore, the LVMI was determined by age (beta = 0.16; *p* = 0.015), the presence of hypertension (beta = 0.15; *p* = 0.01), and additionally by BMI (beta = 0.16; *p* = 0.009) and sex—female (beta = −0.34; *p* < 0.001); R2 = 0.20; *p* < 0.0001 for the model.

## 4. Discussion

### 4.1. Baseline Characteristics of the Studied Population

Our study group included the oldest community-dwelling individuals aged between 77 and 91 years. The frequency of cardiovascular diseases or comorbidities was close to the current epidemiological reports [2,28,29,30]. Resting systolic BP, diastolic BP and heart rate were well-controlled in the studied population.

### 4.2. Dietary Zinc Intake in the Elderly

It is estimated that zinc deficiency may affect a large number of people worldwide and vary between countries, depending on the studied population [31]. For example, in the NHANES III study among people 71+ years of age, insufficient zinc intake (assessed by a 24 hour interview and defined as 77% of the RDA) was found in 66% of males and 68% of females [32]. In our study, 36% of males and 24% of females had inadequate zinc intake as defined by the Federation of American Societies for Experimental Biology [33]. This is better than in the NHANES III study, though still an unsatisfactory result, and is most likely related to the fact that our population included educated people without serious health burdens. Unfortunately, deficiencies are more common in the group of males, whose diet is usually worse [34]. The median consumption per day and per 1000 kcal was similar to the results obtained by other investigators [35]. Higher zinc intake in men is typical as males have an approximately 3 mg higher requirement for this nutrient. The RDA standard for people over 75 years of age adopted by the National Institute of Public Health in Poland is 8 mg for women and 11 mg for men [24]. The main sources of zinc in the diet of the elderly are grain products, vegetables and meat, and for some people the most important source are potatoes—a product with low bioavailability of zinc.

Nevertheless, Dietary Reference Values for zinc for adults vary from country to country and range from 4.2 to 14 mg in men and 3 to 12 mg in women [35]. It depends on the bioavailability of this mineral, among other factors. Zinc absorption is influenced by many factors and it is even recommended that the RDA in the elderly should be increased by nearly one-half of the current values [36]. The average zinc coverage is relatively high, especially among women, however, 46 (64%) men and 95 (50%) women consumed zinc below the Polish level of RDA.

Additionally, it has been observed that with age, the number of milligrams of zinc per 1000 kcal decreases, while within the study group there was no significant difference in the amount of zinc consumed.

### 4.3. Characteristics of the Cardiac Structure and Function of the Studied Population

As for the applicable standards, the results of obtained echocardiographic parameters fell within the range of normal or only slightly abnormal values [25,26]. However, LV mass, LVMI and RWT were different from the norms for the general population, reflecting features of concentric hypertrophy, typical for this age group. The age-related changes of cardiac structure include especially increased LV wall thickness and LV mass [37,38,39,40]. In our study we observed a significant positive correlation between age and LVMI, which confirms the findings of previous reports. Similarly to other reports, LAVI and E/E’ ratio, increasing with age, as determinants of diastolic function deterioration were particularly expressed in the female subgroup of the studied population [28,40]. Furthermore, sex as an additional factor implicates the interpretation of some echocardiographic parameters of cardiac structure and function. Lower chamber dimensions, volumes and LV mass are generally observed in women rather than in men [25,39,41]. This relationship was also noticeable in our study, where lower LA and RV diameter, LV systolic and diastolic volume, as well as LV mass, were recorded in the female subgroup.

A majority of the study population had normal LV systolic function, with median LVEF value at 60%, with no difference observed in relation to sex. The association between age and LV systolic function varies across studies—authors report LVEF to decrease, not to change or to increase with age [38,39,40,42]. We noticed a weak negative correlation with age; this relationship lost statistical significance when analyzed by sex. The number of men (Males) studied in this manuscript was much lower than women (Females), which may lead to the lack of this association between LVEF and zinc intake in men. In the study of Eggers et al. conducted among 70-year-old individuals, LVEF decreased in the next decade of their life, still remaining within the range of normal values, though. There was no reference to sex in this study [43]. Gong et al., in the group of 2358 older adults (median age 71 years) reported baseline LVEF values comparable to ours [38]. The authors observed age-associated longitudinal increase in LVEF during approximately four-year follow-up period, which was more pronounced in women. Nevertheless, in a cross-sectional analysis the above correlations were not present. To conclude, the variety of data on LVEF changes in the elderly indicates that there is a great number of other factors that may influence these relationships. Regardless of age, various diseases cause damage to the heart muscle and its adverse remodeling [44,45,46]. Our results confirm the fact that the strongest independent factor influencing LVEF is the presence of coronary artery disease (*p* < 0.001; β = −0.39), in addition to age (*p* = 0.036; β = −0.12).

Moreover, the structure and function of the heart may be strongly affected by hypertension, which plays a major role in cardiac hypertrophy and diastolic dysfunction. These relationships were also revealed in the results of our research, where only age and hypertension significantly influenced the E/E’ ratio value in multiple stepwise analysis. Similarly, the factors associated with LVMI were age, sex, presence of hypertension and BMI.

### 4.4. Zinc and Cardiac Function

In the context of cardiovascular diseases, zinc plays an invaluable cardio- and vascular-protective role as it is involved in many cellular processes [12,47,48]. Apart from its antioxidative and anti-inflammatory properties, zinc also reduces the activity of caspase-3 and TNF-α, preventing vascular and myocardial cells apoptosis [15,49]. Additionally, experimental studies highlight the significant role of zinc in heart muscle contractility and relaxation via regulation of membrane Ca2+-permeable channels [48].

The most important echocardiographic parameter of LV systolic function is LVEF [2,25]. The results of research evaluating the relationship between zinc and LVEF varied depending on population studied. Frustaci et al. investigated the pathogenesis of dilated cardiomyopathy in 18 patients with intestinal bypass as treatment for severe obesity [14]. The results of endomyocardial biopsy showed severe zinc and selenium deficiency in parallel with pathological changes of cardiomyocytes. Significant increase in LVEF (from 27% to 42%, *p* < 0.001) was observed after six months of Se/Zn infusion treatment as compared to an untreated group, which indicates reversibility of cardiac tissue damage in these patients. Similar findings were published by Rosenblum et al. as a case report of young women with anorexia nervosa and newly developed severe heart failure in whom increase of LVEF from a very low to normal value after zinc supplementation was observed [15]. In our study, in the population without severe deterioration of LV systolic function, a significant correlation between zinc intake and LVEF was found, especially in the female subgroup. In men, the direction of this relationship was analogous; however, statistical significance was not obtained, which may be due to the relatively small number of men included in our study. The multivariate analysis showed that only %RDA for zinc and history of coronary artery disease were factors independently influencing LVEF value. So far there have been only few studies on LVEF and dietary zinc relationship, focused especially on patients with heart failure [50,51]. The results of these reports varied across studies, therefore further analysis in this field is required.

E/E’ ratio is one of the main parameters used in the evaluation of LV diastolic function, next to LVMI, RWT, LAVI and tricuspid regurgitation velocity [2]. We observed a significant inverse correlation between dietary zinc density and E/E’ ratio (r = −0.127, *p* < 0.05) in the studied population. In the analysis by sex, there was a stronger association with E/E’ ratio (r = −0.281, *p* < 0.05), and also a positive correlation with E’ in the men; however, the relationships were not observed in the women. In our results, dietary zinc intake—assessed as %RDA—negatively correlates with LVMI (r = −0.137, *p* < 0.05), which may have a beneficial clinical value for age- or disease-related cardiac remodeling. This finding is consistent with a previous study by Huang et al. who reported a significant inverse relationship between serum zinc levels and LV mass (*p* = 0.029, β = −0.130 for males, *p* = 0.007, β = −0.169 for females) or LVMI (*p* = 0.017, β = −0.142 for males, *p* = 0.004, β = −0.192 for females) in patients with LV hypertrophy [52]. There are only few studies concerning the association of zinc and diastolic function parameters [13,16]. Moreover, researchers have focused mostly on serum zinc levels in patients with impaired cardiac function. In the study by Alexanian et al., serum zinc levels were significantly lower both in patients with acute and chronic heart failure, and negatively correlated with E/E’ ratio (r = −0.349, *p* = 0.001) [13]. Huang et al. studied hemodialyzed patients—a population particularly vulnerable to the development of diastolic dysfunction [16]. They observed an inverse correlation between serum zinc levels and E/E’ ratio (r = −0.217, *p* = 0.003), and LAVI I (r = −0.197, *p* = 0.007), however, not with LV mass or LVMI. Although previous studies as well as our own have examined different populations, all the results suggest a positive role of zinc in the mechanisms associated with diastolic dysfunction, regardless of underlying diseases. However, it is noteworthy that serum zinc levels do not reflect accurately the nutritional status at the individual level.

As recommended, echocardiographic assessment of RV function should include at least one of the RV parameters [2]. TAPSE—as a measurement of basal RV segment displacement—corresponds well to the global RV systolic function [2,53]. Within the normal range, minor sex-related differences were observed. Slightly higher TAPSE values are detected in men as compared to women, and this relationship is also reflected in our results. Additionally, no age-related association was found in the literature, which is also demonstrated by our study on individuals of advanced age [2,54].

What appears to be a new finding of our study is the occurrence of a statistically significant positive correlation between zinc consumption and TAPSE, especially in men. This relationship with RV systolic function was also confirmed according to S’ RV—the parameter determined by another echocardiographic method—that positively correlates with dietary zinc intake (in mg per 1000 kcal) in men. Moreover, daily zinc intake was an independent factor affecting TAPSE value, apart from BMI, female sex, heart rate and use of beta blockers.

### 4.5. Study Limitations

It should be acknowledged that the study was conducted with a group of volunteers representing the Central European population, Caucasians only. Therefore, the results may be different in other cultures. The 24 hour dietary recall questionnaire has some limitations, such as the omission of dietary ingredients or recall bias.

Moreover, echocardiographic assessment did not include measurement of global longitudinal strain (GLS)—a more sensitive determinant of myocardial dysfunction. Furthermore, not all the cardiac parameters could be determined in each participant; this, however, corresponds to everyday practice, where individual body structure may limit image quality. Additionally, we assessed only dietary zinc intake in relation to echocardiographic parameters; other methods of assessing zinc status in the body have not been analyzed.

## 5. Conclusions

To the best of our knowledge, this is the first paper showing a relationship of dietary zinc intake and echocardiographic assessment in older people. In individuals of very advanced age, some parameters of cardiac function are related to zinc intake and a lower level of the micronutrient is associated with poorer heart contractility and relaxation.

Results of correlations and multiple stepwise regressions showed that %RDA of zinc from diet was one of the most important determinants of both LVEF and TAPSE in the studied population. In this regard, the results suggest that adequate dietary zinc intake may improve cardiac function in the elderly, regardless of age- or sex-related cardiac remodeling. Some gender differences were observed, but they require further confirmation.

Given the beneficial effect of zinc on the echocardiographic parameters, as well as the fact that the main dietary sources of zinc for older adults have low bioavailability, increasing the zinc recommendations for this group of people should be considered.

## Figures and Tables

**Figure 1 antioxidants-12-00265-f001:**
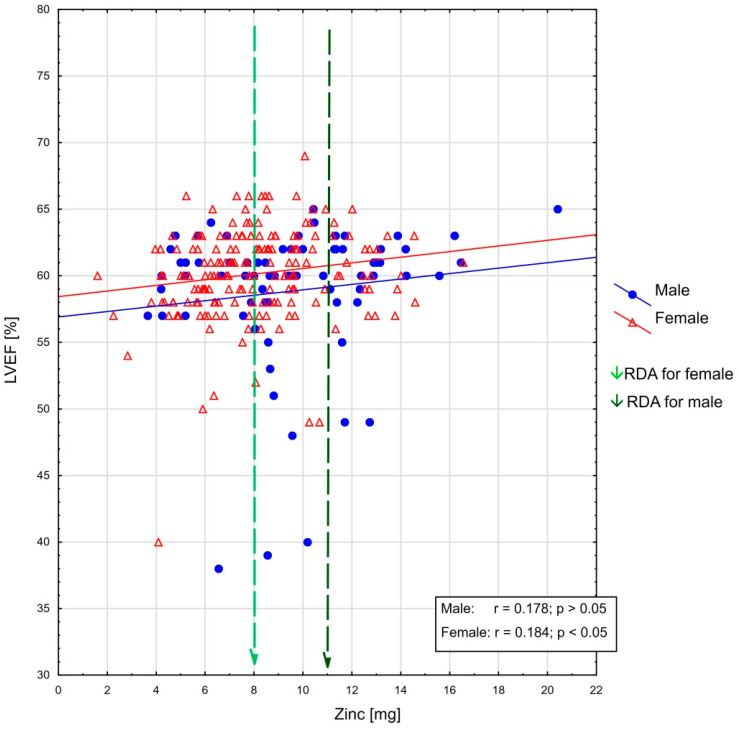
The association between LVEF [%] and zinc intake as well as RDA recommendations in the female and male groups (linear trend line added for easier interpretation).

**Figure 2 antioxidants-12-00265-f002:**
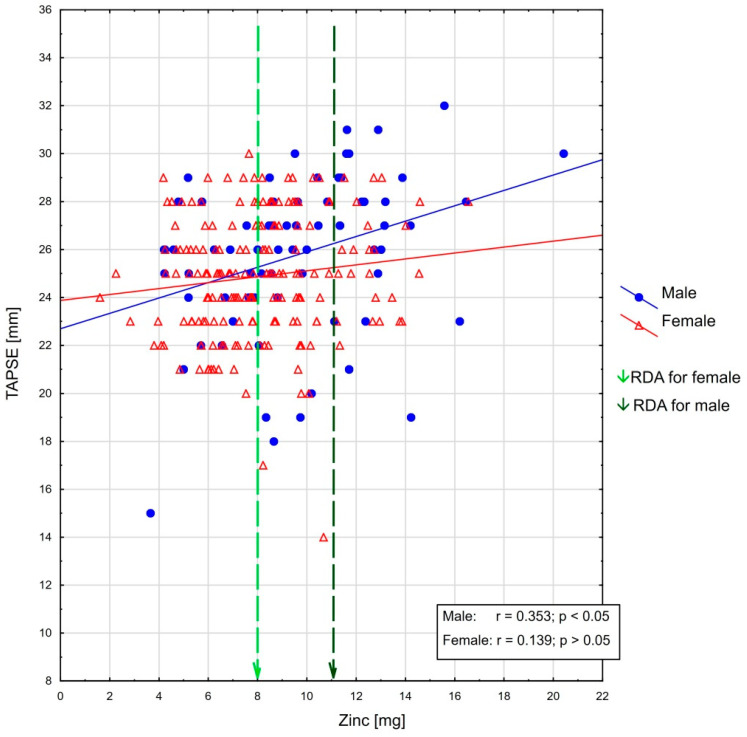
The association between TAPSE [mm] and zinc intake as well as RDA recommendations in the female and male groups (linear trend line added for easier interpretation).

**Table 1 antioxidants-12-00265-t001:** General characteristics of the study population according to sex. The quantitative values are expressed by median, lower and upper quartile or mean and standard deviation, qualitative values as number and percentage.

Variable	All (*n* = 251)	Male (*n* = 71)	Female (*n* = 180)	*p* (Male vs Female)
Age [years]	80 (78–83)	80 (78–84)	80 (78–83)	ns
Education [years]	16 (12–18)	16 (12–18)	16 (12–18)	ns
BMI [kg/m^2^]	27.5 (25.2–30.1)	27.6 (25.9–29.9)	27.4 (24.6–30.2)	ns
WHtR	0.59 (0.54–0.63)	0.59 (0.56–0.63)	0.59 (0.54–0.63)	ns
Systolic blood pressure [mmHg]	135 (122–153)	132 (118–156)	136 (124–152)	ns
Diastolic blood pressure [mmHg]	74 (67–81)	73 (65–81)	75 (68–82)	ns
Heart rate [beats/min]	75 (68–80)	75 (65–85)	75 (68–80)	ns
Zinc [mg]	8.22 (6.31–10.2)	9.42 (7.55–11.7)	7.82 (6.17–9.63)	*p* < 0.001
Zinc [% RDA]	94.8 (74.6–118)	85.7 (68.7–106)	97.7 (77.1–120.4)	0.003
Zinc density [mg/1000 kcal]	5.71 (4.69–6.83)	5.59 (4.68–7.04)	5.74 (4.7–6.78)	ns
Zinc intake ≥ RDA, *n* (%)	109 (43)	25 (35)	84 (47)	ns
Previous stroke; *n* (%)	11 (4.4)	3 (4.2)	8 (4.4)	ns
Previous MI; *n* (%)	13 (5.2)	9 (12.7)	4 (2.2)	*p* < 0.001
Coronary artery disease; *n* (%)	54 (21.5)	26 (36.6)	28 (15.6)	*p* < 0.001
Hypertension; *n* (%)	198 (78.9)	56 (78.9)	142 (79)	ns
Diabetes; *n* (%)	43 (17.1)	13 (18.3)	30 (16.7)	ns
Heart failure; *n* (%)	70 (27.9)	13 (18.3)	57 (31.7)	0.034
COPD; *n* (%)	16 (6.4)	5 (7)	11 (6.1)	ns
Medications; *n* (%):				
ACEI/ARB	158 (63)	47 (66.2)	111 (61.7)	ns
Beta-blockers	154 (61.4)	42 (59.2)	112 (62.2)	ns
Calcium channel blockers	87 (34.7)	22 (31)	65 (36.1)	ns
Diuretics	89 (35.5)	18 (25.4)	71 (39.4)	0.036
MRA	41 (16.3)	14 (19.7)	27 (15)	ns
Statins	147 (58.6)	43 (60.6)	104 (57.8)	ns

ns—not significant. ACEI—angiotensin-converting-enzyme inhibitors. ARB—angiotensin receptor blockers. BMI—body mass index. COPD—chronic obstructive pulmonary disease. MI—myocardial infarction. MRA—mineralocorticoid receptor antagonist. RDA—recommended dietary allowance. WHtR—waist to height ratio.

**Table 2 antioxidants-12-00265-t002:** General characteristics of echocardiographic parameters according to sex.

Variable	All (*n* = 251) Median (Quartiles)	Male (*n* = 71) Median (Quartiles)	Female (*n* = 180) Median (Quartiles)	*p* (Male vs Female)
LA [cm]	3.7 (3.4–4)	4 (3.7–4.2)	3.5 (3.3–3.9)	*p* < 0.001
RVOT [cm]	2.5 (2.3–2.6)	2.7 (2.5–2.8)	2.4 (2.3–2.5)	*p* < 0.001
LAVI [mL/m^2^]	27.08 (22.4–32.5	30.4 (24.4–35.6)	25.8 (21.4–31.6)	*p* < 0.001
LVEDV [mL]	66 (53–80)	83 (73–102)	60 (48–71)	*p* < 0.001
LVEDVI [mL/m^2^]	38.7 (32.6–45.2)	45.3 (40.4–54.1)	35.6 (30.3–42.1)	*p* < 0.001
LVESV [mL]	26 (21–32)	33 (30–40.5)	23 (19–28)	*p* < 0.001
LVESVI [mL/m^2^]	15.3 (12.7–18.1)	18 (15.8–21.9)	14.1 (11.6–16.6)	*p* < 0.001
LVEF [%]	60 (58–62)	60 (58–62)	60 (58–62)	ns
RWT	0.44 (0.41–0.47)	0.44 (0.40–0.47)	0.44 (0.41–0.48)	ns
LV mass [g]	193 (160–227)	237 (203–267)	178 (155–208)	*p* < 0.001
LVMI [g/m^2^]	112 (99–128)	125 (108–142)	108 (95–121)	*p* < 0.001
E [cm/s]	59 (50–71)	59 (50–70)	59.5 (50–72)	ns
E/A ratio	0.7 (0.58–0.81)	0.68 (0.57–0.81)	0.71 (0.58–0.81)	ns
E’ [cm/s]	7 (6–8)	7 (6–8)	7 (6–8)	ns
E/E’ ratio	8.62 (7.28–10.2)	8.40 (7.0–10)	8.64 (7.47–10.3)	ns
S’ [cm/s]	9 (8–9.5)	9 (8–10)	8.5 (8–9)	0.005
TAPSE [mm]	25 (23–27)	26 (24–28)	25 (23–27)	0.007
S’ RV [cm/s]	15 (13–16)	15 (13–16)	15 (13–16)	ns
TR Vmax [m/s]	2.55 (2.3–2.77)	2.6 (2.3–2.77)	2.55 (2.3–2.77)	ns

ns—not significant. E—early mitral diastolic inflow velocity. E/A ratio—ratio of early to late mitral inflow velocities. E/E’ ratio—average ratio of early mitral diastolic inflow velocity to early diastolic mitral annular velocity. E’—average early diastolic mitral annular velocity. LA—left atrial dimension. LAVI—left atrial volume index. LV mass—left ventricular mass. LVEDV—left ventricular end-diastolic volume. LVEDVI—left ventricular end-diastolic volume index. LVEF—left ventricular ejection fraction. LVESV—left ventricular end-systolic volume. LVESVI—left ventricular end-systolic volume index. LVMI—left ventricular mass index. RVOT—right ventricular outflow tract dimension. RWT—relative wall thickness. S’—average systolic mitral annular velocity. S’ RV– systolic right ventricular (tricuspid annular) velocity. TAPSE—tricuspid annular plane systolic excursion. TR Vmax—tricuspid regurgitation velocity.

**Table 3 antioxidants-12-00265-t003:** Spearman’s correlations between echocardiographic parameters, zinc intake and age.

Variable	Zinc [%RDA]	Zinc [mg] /Zinc [%RDA]		Zinc Density [mg/1000 kcal]			Age [Years]		
	All	Female	Male	All	Female	Male	All	Female	Male
LA [cm]	0.000	0.121	0.006	−0.066	−0.136	0.007	0.091	0.048	0.159
RVOT [cm]	0.021	0.133	0.120	0.038	0.046	−0.068	0.057	0.001	0.135
LAVI [mL/m^2^]	−0.024	0.056	−0.091	−0.028	−0.032	−0.072	0.169 **	0.203 **	0.012
LVEDV [mL]	−0.036	0.079	0.134	0.021	0.018	−0.031	0.003	−0.027	−0.105
LVEDVI [mL/m^2^]	−0.038	0.035	0.124	0.016	0.004	−0.013	0.029	0.035	−0.119
LVESV [mL]	−0.063	0.055	0.076	0.020	0.015	−0.013	0.029	0.005	−0.049
LVESVI [mL/m^2^]	−0.072	0.004	0.050	0.015	0.005	−0.010	0.055	0.064	−0.077
LVEF [%]	0.196 *	0.184 *	0.178	0.069	0.047	0.115	−0.147 *	−0.137	−0.156
RWT	−0.072	−0.046	−0.153	−0.028	−0.026	−0.021	0.085	0.058	0.159
LV mass [g]	−0.087	0.000	0.074	0.011	−0.029	−0.003	0.087	0.036	0.184
LVMI [g/m^2^]	−0.137 *	−0.086	−0.010	−0.047	−0.066	−0.038	0.129 *	0.105	0.181
E [cm/s]	0.089	0.086	0.062	0.029	0.054	−0.012	0.063	0.055	0.109
E/A ratio	0.086	0.044	0.138	0.006	0.006	0.018	−0.048	−0.051	−0.044
E’ [cm/s]	0.083	0.033	0.216	0.122	0.040	0.324 **	−0.114	−0.145	−0.034
E/E’ ratio	−0.074	−0.049	−0.189	−0.127 *	−0.052	−0.281 *	0.191 **	0.230 **	0.106
S’ [cm/s]	0.079	0.133	0.073	0.095	0.048	0.205	−0.041	−0.028	−0.098
TAPSE [mm]	0.153 *	0.139	0.353 *	0.021	0.033	−0.027	−0.005	−0.062	0.113
S’ RV [cm/s]	0.046	0.032	0.096	0.078	−0.015	0.289 *	0.089	0.063	0.139
TR Vmax [m/s]	−0.075	−0.059	−0.175	−0.117	−0.051	−0.431	0.139	0.136	0.115
Age [years]	−0.115	−0.120	−0.075	−0.140 *	−0.200 **	0.005	x	x	x

* *p* < 0.05. ** *p* < 0.01. E—early mitral diastolic inflow velocity. E/A ratio—ratio of early to late mitral inflow velocities. E/E’ ratio—average ratio of early mitral diastolic inflow velocity to early diastolic mitral annular velocity. E’—average early diastolic mitral annular velocity. LA—left atrial dimension. LAVI—left atrial volume index. LV mass—left ventricular mass. LVEDV—left ventricular end-diastolic volume. LVEDVI—left ventricular end-diastolic volume index. LVEF—left ventricular ejection fraction. LVESV—left ventricular end-systolic volume. LVESVI—left ventricular end-systolic volume index. LVMI—left ventricular mass index. RDA—recommended dietary allowance. RVOT—right ventricular outflow tract dimension. RWT—relative wall thickness. S’—average systolic mitral annular velocity. S’ RV–systolic right ventricular (tricuspid annular) velocity. TAPSE—tricuspid annular plane systolic excursion. TR Vmax—tricuspid regurgitation velocity.

## Data Availability

All of the data is contained within the article.

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
