# Peer review of "Dietary Zinc Is Associated with Cardiac Function in the Older Adult Population"

_antioxidants, 2023, doi:10.3390/antiox12020265_

Round 1

Reviewer 1 Report (Previous Reviewer 2)

The Authors have addressed all my remarks in the revision. Therefore, I find this work to be sufficient for publication in the form as it is.

Author Response

Thank you very much for all your very valuable comments and time.

Reviewer 2 Report (New Reviewer)

There are very interesting aspects to this manuscript while it presented novelty by revealing the relationship between dietary zinc intake and cardiac structure and function in individuals of advanced age. The only concern is the number of men (Males) studied in this manuscript was much lower than women (females), which may lead to the lack of association between LVEF [%] and zinc intake in men. A paragraph could be added in the Discussion Section to discuss the probable effect of gender on zinc intake and cardiac disfunction among the older adult population.

Author Response

Thank you very much for the very valuable review. Consecutive community-dwelling patients were included in the study. Therefore the differences in the numbers of women and their age in relation to the men reflect the demographic situation in Poland. We also agreed that the lack of statistically significant association between LVEF [%] and zinc intake in men could be connected to a lower population. Therefore, the text of the manuscript was supplemented with the following information:

The difference in the numbers of women (n=180) in relation to the men (n=71) reflects the demographic situation in Poland. In 2019 the sex index for seniors 75+ was 0.51 [27].

  1. Statistics Poland. Information on the situation of the elderly in Poland in 2019. 2020, 585; https://das.mpips.gov.pl/source/2020/Informacja%20za%202019%20r.%2027.10.2020%20r..pdf (access 2nd, January 2023).

The number of men (Males) studied in this manuscript was much lower than women (Females), which may lead to the lack of this association between LVEF and zinc intake in men.

Reviewer 3 Report (New Reviewer)

Carefully performed work, especially the statistical analysis

Author Response

Thank you very much; we really appreciate your review.

Reviewer 4 Report (New Reviewer)

This study presents interesting results of the association between dietary zinc intake and cardiac parameters. The manuscript is well organized, all analyses seem to be appropriately performed, the conclusions are supported by the data presented.  The study has some issues that need to be addressed:

1)  As some dietary supplements may contain zinc, the information should be provided if patients taking such products were included or excluded in the study.

2)  Author should explain the high disproportion in numbers of men and women (71 versus 180) included in the study.

3)  It is intriguing that the authors of this manuscript put forward the hypothesis that zinc intake could be associated with cardiac function. It is quite remarkable that significant associations were found. The explanation will be useful regarding mechanism of action at which zinc may influence the cardiac function parameters including LVEF and TAPSE.

Author Response

Reviewer 4

Comments and Suggestions for Authors

This study presents interesting results of the association between dietary zinc intake and cardiac parameters. The manuscript is well organized, all analyses seem to be appropriately performed, the conclusions are supported by the data presented.  The study has some issues that need to be addressed:

1)  As some dietary supplements may contain zinc, the information should be provided if patients taking such products were included or excluded in the study.

Thank you for pointing out this important issue. One of the criteria for exclusion from the study was the use of zinc supplementation. The text was supplemented with the information.

The exclusion criteria were acute or uncontrolled chronic inflammatory process, end-stage chronic or neoplastic disease, hemodialysis therapy, mental or functional inability to participate in the study, the use of zinc supplementation and lack of patient consent.

2)  Author should explain the high disproportion in the numbers of men and women (71 versus 180) included in the study.

Consecutive community-dwelling patients were included in the study. Therefore the differences in the numbers of women and their age in relation to the men reflect the demographic situation in Poland. In 2019 the sex index for seniors in Poland was:  0.51. The text of the manuscript was supplemented with the following information:

The difference in the numbers of women (n=180) in relation to the men (n=71) reflects the demographic situation in Poland. In 2019 the sex index for seniors 75+ was 0.51 [27].

  1. Statistics Poland. Information on the situation of the elderly in Poland in 2019. 2020, 585; https://das.mpips.gov.pl/source/2020/Informacja%20za%202019%20r.%2027.10.2020%20r..pdf (access 2nd, January 2023).

The number of men (Males) studied in this manuscript was much lower than women (Females), which may lead to the lack of this association between LVEF and zinc intake in men.

3)  It is intriguing that the authors of this manuscript put forward the hypothesis that zinc intake could be associated with cardiac function. It is quite remarkable that significant associations were found. The explanation will be useful regarding the mechanism of action at which zinc may influence the cardiac function parameters including LVEF and TAPSE.

Thank you very much for the very valuable comment. Zinc plays an important role in the function of the heart muscle, being involved in many processes on a cellular level. As was proved before, Zinc has anti-inflammatory properties preventing cardiac damage. On the other hand, Zinc reduces the activity of caspase3 and TNFalpha, which promote cardiac cell apoptosis. As the cardiac cycle consists of the systolic and diastolic phases, cardiomyocyte damage may be clinically seen in the deterioration of systolic function, both of the left and right ventricles, as observed in cardiomyopathies or myocarditis. Echocardiographic evaluation allows diagnosis and assessment of the severity of LV or RV systolic dysfunction, where LVEF and TAPSE are widely used to describe it. The relationship between cardiomyocyte contractility and Zinc may be also due to Zinc regulation of ca2plus membrane channels involving membrane depolarization.

The diastolic properties of the myocardium may deteriorate in the course of aging and various disease, especially hypertension with concomitant cardiac hypertrophy, as was demonstrated in our result. The role of zinc was here not so significant, however, some correlations were observed. it is worth noting that our population was characterized by quite Good diastolic function. The potential mechanism between zinc deficiency and the development of diastolic dysfunction may be the role of zinc in the regulation of the RAA system (renin-angiotensin-aldosterone).

The literature that addresses the various potential effects of zinc on the heart and the relationship with echo parameters and mechanisms is included in the paper, item number 8, 9, 10, 11, 12, 13, 14, 15, 16, 47, 48, 49.

This manuscript is a resubmission of an earlier submission. The following is a list of the peer review reports and author responses from that submission.

Round 1

Reviewer 1 Report

The submitted paper of Szadkowska analysed the relationship between dietary zinc content assessed from questionnaires and cardiac function in older adult population.

Major objectives:

1.       There were some previous studies in this area which were more objective. In particular the assessment of zinc levels simply from food interview is decreasing the objectivity (for example, authors are alone stating at r. 289 – that there are many factors influencing zinc absorption). Other studies measured directly plasma levels of zinc or indirectly assessed zinc status by determining the activity of zinc- containing enzymes (Association of Zinc and Copper Status with Cardiovascular Diseases and their Assessment Methods: A Review Study, e.g. doi: 10.2174/1389557520666200729160416.). Numerically, the correlation coefficients reached only low numbers in this study. This is logical since there were only low differences in food zinc content among enrolled subjects (Table 1), which in addition was very closely related to recommended dietary zinc allowances.  In fact, there was no deficiency in the studied population which lowers the potential usefulness of the obtained outcomes. The low predictive value is also apparent from Table 3 – some correlations were observed only in women, etc.

2.       Scientifically the analysis was not correctly performed as authors used Spearman coefficients but the relationship is linear according them (Figures 1 and 2).

This paper has some merit, but due to these crucial limitations, it has not achieved high quality needed for papers intended to be published in Antioxidants.

Other issue:

I am not sure if there is an expert on cardiovascular system (a cardiologist or cardiovascular pharmacologist) in the author´s team as some parts are written in rather lay and cardiologically unprecise style. Who was responsible for echocardiography? This should be specified in author contribution.

Some specific parts:

r. 40 - Age-related cardiac remodeling involves both cardiomyocytes and the extracellular matrix, which is typically manifested as concentric hypertrophy. concentric hypertrophy can be simply the results of elevated high pressure /must not be directly related to age/ and hypertension was very common in the population tested in this study

 r. 61 coronary heart disease, this is the synonymous term for coronary artery disease, so only one term should be used

 r. 61 Zinc is a catalyst for many enzymes Zinc is found in active centers of many enzymes

 r. 70 - The role of zinc as an essential microelement that affects myocardial and vascular properties in multiple physiologic pathways has been previously emphasized rewritting is needed

 r. 68 systolic and diastolic disorders -  rewritting is needed

 r. 94 -  The level of education is presented as the number of years of education. this is not very suitable, it would be better to classify it as the highest achieved education (doctoral, Master-Bachelor, secondary or primary school)

Author Response

  1. There were some previous studies in this area which were more objective. In particular the assessment of zinc levels simply from food interview is decreasing the objectivity (for example, authors are alone stating at r. 289 – that there are many factors influencing zinc absorption). Other studies measured directly plasma levels of zinc or indirectly assessed zinc status by determining the activity of zinc- containing enzymes (Association of Zinc and Copper Status with Cardiovascular Diseases and their Assessment Methods: A Review Study, e.g. doi: 10.2174/1389557520666200729160416.).

Authors’ answer:

We thank the Reviewer for drawing our attention to other than our methods of assessing zinc metabolism in the body. However, we aimed to assess the exogenous supply of zinc and the potential relationship between this supply and cardiovascular function. The special issue for which we have prepared this article is about antioxidants in the diet ("Dietary Antioxidants and Cardiovascular Health".)

The submitted article allowed us to take a broader look at the methods of assessing the zinc status and, as emphasized in this mini-review, the assessment of intake with a zinc diet is one of the important methods that can be used. We quoted this article in the Introduction.

Therefore, it is important to analyze the body's zinc status, which can be assessed by various methods, e.g. plasma levels of zinc, the activity of zinc-containing enzymes, and dietary zinc intake [17].

  1. Malekahmadi, M.; Firouzi, S.; Rezayi, M.; Ghazizadeh, H.; Ranjbar, G.; Ferns, G.A.; Mobarhan, M.G. Association of Zinc and Copper Status with Cardiovascular Diseases and their Assessment Methods: A Review Study. Mini Rev Med Chem 2020, 20, 2067-2078, doi:10.2174/1389557520666200729160416.

We are aware of the strengths and weaknesses of the 24-h dietary recall, but we wanted to use a simple, cheap, and widely used method in everyday clinical practice. In our analysis, we referred to consumption norms, which are the main population recommendation. These standards assume a certain nutritional quality (biological value) of food, which determines the availability of its components to the body. Work on the development of standards in the European Union is dealt with by the EFSA Panel on Dietetic Products, Nutrition and Allergies (NDA). In order to ensure a consistent approach to the development of standards, the Panel in 2010 established general principles for work in this area (EFSA Panel on Dietetic Products, Nutrition, and Allergies (NDA), Scientific opinion on principles for deriving and applying Dietary Reference Values, EFSA Journal, 2010, 8, 3, 1458).

All EFSA draft Dietary Reference Values (DRVs) opinions are subject to public consultation with Member States, the scientific community and other stakeholders before finalization. This ensures that EFSA uses the widest range of opinions to gather the most up-to-date, precise and comprehensive information. For the population we studied, official standards updated in 2017 were used.

The method used by us is also commonly used by other researchers. References presenting works using this method have been added to the methodology:

2.2. Zinc intake

The level of zinc consumption was assessed using a modified 24-hour interview method [18,19].

  1. Yao, J.; Hu, P.; Zhang, D. Associations Between Copper and Zinc and Risk of Hypertension in US Adults. Biol Trace Elem Res 2018, 186, 346-353, doi:10.1007/s12011-018-1320-3.
  2. Yao, B.; Wang, Y.; Xu, L.; Lu, X.; Qu, H.; Zhou, H. Associations Between Copper and Zinc and High Blood Pressure in Children and Adolescents Aged 8-17 Years: an Exposure-Response Analysis of NHANES 2007-2016. Biol Trace Elem Res 2020, 198, 423-429, doi:10.1007/s12011-020-02095-x.

  1. Numerically, the correlation coefficients reached only low numbers in this study. This is logical since there were only low differences in food zinc content among enrolled subjects (Table 1), which in addition was very closely related to recommended dietary zinc allowances.  In fact, there was no deficiency in the studied population which lowers the potential usefulness of the obtained outcomes. The low predictive value is also apparent from Table 3 – some correlations were observed only in women, etc.

Authors’ answer:

The observed relationships between the supply of zinc in the diet and cardiac parameters are relatively low because many other factors may affect these relationships, e.g. the occurrence of hypertension, coronary artery disease, or age, which came out in our research and was also emphasized in the discussion. We would also like to point out that correlations with serum zinc and echocardiographic parameters found by other authors are also relatively low (these values were added in the discussion). Our study also points to possible gender differences. Both the intake of zinc in milligrams and the % of the norm are statistically significantly different in the group of women and men (see Table 1). In the analysis of consumption, the presented differences are large and significant also from the clinical point of view. Extremely malnourished people and people with severe circulatory failure or infections were deliberately not included in the analysis, and the homogeneity of the group is its strength.

In order to better show the Reviewer and readers what number of people consumed the right amount of zinc, green vertical arrows representing the RDA level for women (8 mg) and men (11 mg) were added to graphs 1 and 2, respectively. Those to the right of the arrow have an adequate intake those to the left of the arrow have an intake below the RDA.

Figures titles changed:

Figure 1. The association between LVEF [%] and zinc intake as well as RDA recommendations in the female and male groups (linear trend line added for easier interpretation).

Figure 2. The association between TAPSE [mm] and zinc intake as well as RDA recommendations in the female and male groups (linear trend line added for easier interpretation). 

It is known that aging processes, also in relation to the myocardium, proceed differently depending on gender, which was taken into account in the study. Of course, these interesting differences, included in Table 3, require further analysis. We plan to perform in the future complementary analyzes using the indicated methods of full zinc status assessment.

  1. Conclusions

To the best of our knowledge, this is the first paper showing a relationship of dietary zinc intake and echocardiographic assessment in older people. In individuals of very advanced age, some parameters of cardiac function are related to zinc intake and a lower level of the micronutrient is associated with poorer heart contractility and relaxation.

Results of correlations and multiple stepwise regressions showed that %RDA of zinc from diet was one of the most important determinants of both LVEF and TAPSE in the studied population. In this regard, the results suggest that adequate dietary zinc intake may improve cardiac function in the elderly, regardless of age- or sex-related cardiac remodeling. Some gender differences were observed, but they require further confirmation.

Given the beneficial effect of zinc on the echocardiographic parameters, as well as the fact that the main dietary sources of zinc for older adults have low bioavailability, increasing the zinc recommendations for this group of people should be considered.

  1. Scientifically the analysis was not correctly performed as authors used Spearman coefficients but the relationship is linear according them (Figures 1 and 2).

This paper has some merit, but due to these crucial limitations, it has not achieved high quality needed for papers intended to be published in Antioxidants.

Authors’ answer:

Spearman correlation was used to confirm significance. For graphical representation and clarity of interpretation, we added a linear trend line as there is no meaningful way of representing rank correlation that would be easy to interpret. We added information about this to the figure legend. We performed additional analyzes to check the associations with the Pearson correlation coefficients and we obtained comparable results.

The normality of distribution was verified using the Shapiro-Wilk test. Variables without a normal distribution were presented as median value and quartile range (from 25% to 75% quartile), with a normal distribution as median and standard deviations. Between groups, the quantitative variables were compared using the one-way ANOVA or Mann-Whitney U-test and the qualitative variables using the chi-square test or Fischer’s Exact test. The correlations between the numerical data were analyzed through Spearman‘s correlation coefficient.

Figure 1. The association between LVEF [%] and zinc intake as well as RDA recommendations in the female and male groups (linear trend line added for easier interpretation).

Figure 2. The association between TAPSE [mm] and zinc intake as well as RDA recommendations in the female and male groups (linear trend line added for easier interpretation). 

Other issue:

I am not sure if there is an expert on cardiovascular system (a cardiologist or cardiovascular pharmacologist) in the author´s team as some parts are written in rather lay and cardiologically unprecise style. Who was responsible for echocardiography? This should be specified in author contribution.

Authors’ answer:

Thank you for pointing out the need to clarify the terms used. The team includes a specialist cardiologist who performed the echocardiography examination and prepared the manuscript. Echocardiography was included in Contributions, and appropriate information was added to the methodology:

Author Contributions: Conceptualization, I.S., A.G., R.W.; methodology, A.G., I.S.; echocardiography, I.S.; software, A.G.; validation, I.S. and A.G.; formal analysis, A.G; data curation, A.G., I.S., R.W.,LK; writing—original draft preparation, A.G., I.S., R.W.; visualization, A.G.; supervision, A.G., T.K., A.J.; project administration, T.K.; funding acquisition, T.K. All the authors have read and agreed to the published version of the manuscript.

Transthoracic echocardiography was performed by the same investigator cardiologist using a Vivid S70 ultrasound system (GE Medical Systems, 2018).

Some specific parts:

  1. 40 - Age-related cardiac remodeling involves both cardiomyocytes and the extracellular matrix, which is typically manifested as concentric hypertrophy. – concentric hypertrophy can be simply the results of elevated high pressure /must not be directly related to age/ and hypertension was very common in the population tested in this study

Authors’ answer:

The sentence concerned age-related remodeling of the heart, independent of other factors. For a better presentation of the described relationships and to avoid misunderstanding, the part of the text related to age-related changes has been reduced and the discussion has been extended to include comorbidities. We supplemented the results with regression analysis, taking into account hypertension as a potential factor affecting the LVMI and E/E' ratio.

As Reviewers noted, our results confirm that hypertension and age have the greatest impact on the myocardial mass and diastolic function, regardless of other diseases and zinc intake.

Regardless of age, various diseases cause damage to the heart muscle and its adverse remodeling [43-45]. Our results confirm the fact that the strongest independent factor influencing LVEF is the presence of coronary artery disease (p<0.001; β= -0.39), in addition to age (p=0.036; β= -0.12).

Moreover, the structure and function of the heart may be strongly affected by hypertension, which plays a major role in cardiac hypertrophy and diastolic dysfunction. These relationships were also revealed in the results of our research, where only age and hypertension significantly influenced the E/E’ ratio value in multiple stepwise analysis. Similarly, the factors associated with LVMI were age, sex, presence of hypertension and BMI.

  1. González, A.; Ravassa, S.; López, B.; Moreno, M.U.; Beaumont, J.; San José, G.; Querejeta, R.; Bayés-Genís, A.; Díez, J. Myocardial Remodeling in Hypertension. Hypertension 2018, 72, 549-558, doi:10.1161/hypertensionaha.118.11125.
  2. Jia, G.; Hill, M.A.; Sowers, J.R. Diabetic Cardiomyopathy: An Update of Mechanisms Contributing to This Clinical Entity. Circ Res 2018, 122, 624-638, doi:10.1161/circresaha.117.311586.
  3. Kemp, C.D.; Conte, J.V. The pathophysiology of heart failure. Cardiovasc Pathol 2012, 21, 365-371, doi:10.1016/j.carpath.2011.11.007.

  1. 61 – coronary heart disease, this is the synonymous term for coronary artery disease, so only one term should be used

Authors’ answer:

For uniform nomenclature, "coronary heart disease" has been changed to "coronary artery disease".

It has also been reported that low zinc intake in men is associated with higher mortality from coronary artery disease [6].

  1. 61 – Zinc is a catalyst for many enzymes – Zinc is found in active centers of many enzymes…

Authors’ answer:

Thank you for this comment. The text was corrected as recommended by the Reviewer:

Zinc is found in active centers of many enzymes and proteins and ensures the efficiency of the immune system.

  1. 70 - The role of zinc as an essential microelement that affects myocardial and vascular properties in multiple physiologic pathways has been previously emphasized – rewritting is needed

Authors’ answer:

Thank you for this comment. The text was corrected as recommended by the Reviewer:

From a biochemical point of view, zinc is one of the microelements essential for the proper functioning of the cardiovascular system.

  1. 68 – systolic and diastolic disorders -  rewritting is needed

Authors’ answer:

As suggested by the Reviewer: disorders changed to dysfunction:

  1. 94 -  The level of education is presented as the number of years of education. – this is not very suitable, it would be better to classify it as the highest achieved education (doctoral, Master-Bachelor, secondary or primary school)

Authors’ answer:

Since the surveyed group consisted of people born before and during World War II, their education was non-standard, and the highest diplomas received did not always correspond to the traditional learning process. In order to avoid errors in interpretation, the method of evaluating education based on the number of years of education was used. This method is widely used. Examples below:

  • Langa KM, Larson EB, Crimmins EM, Faul JD, Levine DA, Kabeto MU, Weir DR. A Comparison of the Prevalence of Dementia in the United States in 2000 and 2012. JAMA Intern Med. 2017 Jan 1;177(1):51-58. doi: 10.1001/jamainternmed.2016.6807. PMID: 27893041; PMCID: PMC5195883.
  • Turney IC, Lao PJ, Rentería MA, Igwe KC, Berroa J, Rivera A, Benavides A, Morales CD, Rizvi B, Schupf N, Mayeux R, Manly JJ, Brickman AM. Brain Aging Among Racially and Ethnically Diverse Middle-Aged and Older Adults. JAMA Neurol. 2022 Nov 14. doi: 10.1001/jamaneurol.2022.3919. Epub ahead of print. PMID: 36374494.
  • Marbaniang SP, Patel R, Kumar P, Chauhan S, Srivastava S. Hearing and vision difficulty and sequential treatment among older adults in India. Sci Rep. 2022 Nov 9;12(1):19056. doi: 10.1038/s41598-022-21467-y. PMID: 36351946; PMCID: PMC9646738.
  • Reinhart RMG, Nguyen JA. Working memory revived in older adults by synchronizing rhythmic brain circuits. Nat Neurosci. 2019 May;22(5):820-827. doi: 10.1038/s41593-019-0371-x. Epub 2019 Apr 8. PMID: 30962628; PMCID: PMC6486414.

We are convinced that thanks to the reviewer's comments, we have adequately supplemented the manuscript. New analyzes were added, the impact of comorbidities included, and RDA values included in graphs. We hope that in its current form the article is properly prepared for the special issue: Special Issue "Dietary Antioxidants and Cardiovascular Health".

Reviewer 2 Report

The work “Dietary zinc is associated with cardiac function in the older adult population“ by Iwona Szadkowska, Agnieszka Guligowska, Rafał Nikodem Wlazeł, Łukasz Kroc, Anna Jegier, and Tomasz Kostka touches an important topic of the zinc intake and cardiac function.

The work is well written and presents a good set of data.

However, I have a few remarks:

There are several too generous statements, for instance, line 276: It is estimated that zinc deficiency may affect a large number of people – from 4% to 73% worldwide - depending on the studied population [32]. The range 4-73% has no informative value.

The work show correlation between zinc intake and cardiovascular diseases. However, it does not necceserly transfer to the cause and effect relationship.

Zinc intake is based on 24-hour interviews. Thus, zinc intake was considered very generally, without considering the specific influence of other food products on zinc absorption.

I would expect to Conclusion section to be more precise and widely elaborated.

Recommendation: Major revision which addresses the abovementioned points.

Author Response

The work “Dietary zinc is associated with cardiac function in the older adult population“ by Iwona Szadkowska, Agnieszka Guligowska, Rafał Nikodem Wlazeł, Łukasz Kroc, Anna Jegier, and Tomasz Kostka touches an important topic of the zinc intake and cardiac function.

The work is well written and presents a good set of data.

However, I have a few remarks:

There are several too generous statements, for instance, line 276: It is estimated that zinc deficiency may affect a large number of people – from 4% to 73% worldwide - depending on the studied population [32]. The range 4-73% has no informative value.

Authors’ answer:

The range to which we have referred is presented in [Prasad, A.S. Discovery of human zinc deficiency: 50 years later. J Trace Elem Med Biol 2012, 26, 66-69, doi:10.1016/j.jtemb.2012.04.004.]. The problem of zinc deficiency, as well as the level of intake standards is widely discussed, and we tried to present the scale of the problem as objectively as possible. The text was corrected and supplemented:

It is estimated that zinc deficiency may affect a large number of people worldwide and vary between countries - depending on the studied population [30]. For example, in the NHANES III study among people 71+ years of age, insufficient zinc intake (assessed by a 24-hour interview and defined as 77% of the RDA) was found in 66% of males and 68% of females [31]. In our study, as defined by the Federation of American Societies for Experimental Biology [32], 36% of males and 24% of females had inadequate zinc intake. This better that in the NHANES III study, though still unsatisfactory result, is most likely related to the fact that our population included educated people without serious health burdens. Unfortunately, deficiencies are more common in the group of males, whose diet is usually worse [33].

  1. Prasad, A.S. Discovery of human zinc deficiency: 50 years later. J Trace Elem Med Biol 2012, 26, 66-69, doi:10.1016/j.jtemb.2012.04.004.
  2. Briefel, R.R.; Bialostosky, K.; Kennedy-Stephenson, J.; McDowell, M.A.; Ervin, R.B.; Wright, J.D. Zinc intake of the U.S. population: findings from the third National Health and Nutrition Examination Survey, 1988-1994. J Nutr 2000, 130, 1367s-1373s, doi:10.1093/jn/130.5.1367S.
  3. Office, F.o.A.S.f.E.B.L.S.R.; Monitoring, I.B.f.N.; Research, R. Third report on nutrition monitoring in the United States; Interagency Board for Nutrition Monitoring: 1995; Volume 1.
  4. Tay, E.; Barnett, D.; Leilua, E.; Kerse, N.; Rowland, M.; Rolleston, A.; Waters, D.L.; Edlin, R.; Connolly, M.; Hale, L.; et al. The Diet Quality and Nutrition Inadequacy of Pre-Frail Older Adults in New Zealand. Nutrients 2021, 13, doi:10.3390/nu13072384.

The work show correlation between zinc intake and cardiovascular diseases. However, it does not necceserly transfer to the cause and effect relationship.

Authors’ answer:

As the Reviewer rightly pointed out, no causes were investigated; the study was an observational study (without intervention). This issue has been clearly stated in the limitations of the study. People with significant heart damage, indicating an advanced disease process, were excluded from our research group. And the presented study itself is a preliminary result for planning an intervention study that will show cause and effect relationships.

4.5. Study limitations 

It should be acknowledged that the study was conducted with a group of volunteers representing the Central-European population, Caucasians only. Therefore, the results may be different in other cultures. The 24-hour dietary recall questionnaire has some limitations, such as the omission of dietary ingredients or recall bias.

Moreover, echocardiographic assessment did not include measurement of global longitudinal strain (GLS) - a more sensitive determinant of myocardial dysfunction. Furthermore, not all the cardiac parameters could be determined in each participant, what however, corresponds to everyday practice, where individual body structure may limit image quality. Additionally, we assessed only dietary zinc intake in relation to echocardiographic parameters, other methods of assessing zinc status in the body have not been analyzed.

Zinc intake is based on 24-hour interviews. Thus, zinc intake was considered very generally, without considering the specific influence of other food products on zinc absorption.

Authors’ answer: -

Thank you for drawing our attention to other than our methods of assessing zinc metabolism in the body. However, our aim was to assess the exogenous supply of zinc and the potential relationship between this supply and cardiovascular function.

The submitted article allowed us to take a broader look at the methods of assessing the zinc economy and, as emphasized in this work, the assessment of intake with a zinc diet is one of the very important methods that can be used. We quoted this article in the Introduction.

Therefore, it is important to analyze the body's zinc status, which can be assessed by various methods, e.g. plasma levels of zinc, the activity of zinc-containing enzymes, and dietary zinc intake [17].

  1. Malekahmadi, M.; Firouzi, S.; Rezayi, M.; Ghazizadeh, H.; Ranjbar, G.; Ferns, G.A.; Mobarhan, M.G. Association of Zinc and Copper Status with Cardiovascular Diseases and their Assessment Methods: A Review Study. Mini Rev Med Chem 2020, 20, 2067-2078, doi:10.2174/1389557520666200729160416.

We are aware of the strengths and weaknesses of the 24-h dietary recall, but we wanted to use a simple, cheap and widely used method in everyday clinical practice. In our analysis, we referred to consumption norms, which are the main population recommendation. These standards assume a certain nutritional quality (biological value) of food, which determines the availability of its components to the body. Work on the development of standards in the European Union is dealt with by the EFSA Panel on Dietetic Products, Nutrition and Allergies (NDA). In order to ensure a consistent approach to the development of standards, the Panel in 2010 established general principles for work in this area (EFSA Panel on Dietetic Products, Nutrition, and Allergies (NDA), Scientific opinion on principles for deriving and applying Dietary Reference Values, EFSA Journal, 2010, 8, 3, 1458).

All EFSA draft Dietary Reference Values (DRVs) opinions are subject to public consultation with Member States, the scientific community and other stakeholders before finalization. This ensures that EFSA uses the widest range of opinions to gather the most up-to-date, precise and comprehensive information. For the population we studied, official standards updated in 2017 were used.

The method used by us is also commonly used by other researchers. References presenting works using this method have been added to the methodology:

2.2. Zinc intake

The level of zinc consumption was assessed using a modified 24-hour interview method [18,19].

  1. Yao, J.; Hu, P.; Zhang, D. Associations Between Copper and Zinc and Risk of Hypertension in US Adults. Biol Trace Elem Res 2018, 186, 346-353, doi:10.1007/s12011-018-1320-3.
  2. Yao, B.; Wang, Y.; Xu, L.; Lu, X.; Qu, H.; Zhou, H. Associations Between Copper and Zinc and High Blood Pressure in Children and Adolescents Aged 8-17 Years: an Exposure-Response Analysis of NHANES 2007-2016. Biol Trace Elem Res 2020, 198, 423-429, doi:10.1007/s12011-020-02095-x.

I would expect to Conclusion section to be more precise and widely elaborated.

 Authors’ answer:

As suggested by the reviewer, the conclusions were extended:

  1. Conclusions

To the best of our knowledge, this is the first paper showing a relationship of dietary zinc intake and echocardiographic assessment in older people. In individuals of very advanced age, some parameters of cardiac function are related to zinc intake and a lower level of the micronutrient is associated with poorer heart contractility and relaxation.

Results of correlations and multiple stepwise regressions showed that %RDA of zinc from diet was one of the most important determinants of both LVEF and TAPSE in the studied population. In this regard, the results suggest that adequate dietary zinc intake may improve cardiac function in the elderly, regardless of age- or sex-related cardiac remodeling. Some gender differences were observed, but they require further confirmation.

Given the beneficial effect of zinc on the echocardiographic parameters, as well as the fact that the main dietary sources of zinc for older adults have low bioavailability, increasing the zinc recommendations for this group of people should be considered.

Recommendation: Major revision which addresses the abovementioned points.

We are convinced that thanks to the reviewer's comments, we have adequately supplemented the manuscript. New analyzes were added, the impact of comorbidities included, and RDA values included in graphs. We hope that in its current form the article is properly prepared for the special issue: Special Issue "Dietary Antioxidants and Cardiovascular Health".